



# Sea-level and monsoonal control on the Maldives carbonate platform (Indian Ocean) over the last 1.3 million years

Montserrat Alonso-Garcia [1,2], Jesus Reolid [3], Francisco J. Jimenez-Espejo [4], Or M. Bialik [5], Carlos A. Alvarez Zarikian [6], Juan Carlos Laya [7], Igor Carrasquiera[8], Luigi Jovane [8], John J.G. Reijmer [9], Christian Betzler [10], Gregor P. Eberli [11]

[1] Dpto. Geologia, Universidad de Salamanca, Pza de los caídos s/n, 37008 Salamanca, Spain
[2] Centro de Ciencias do Mar (CCMAR), Universidade do Algarve, Campus de Gambelas, 8005-139 Faro, Portugal
[3] Departamento de Estratigrafía y Paleontología, Facultad de Ciencias, Universidad de Granada, Avenida de la Fuente Nueva S/N, 18071 Granada, Spain
[4] Instituto Andaluz de Ciencias de la Tierra (CSIC-UGR), Avda. de las Palmeras 4, 18100, Armilla, Spain
[5] Dr. Moses Strauss Department of Marine Geosciences, Charney School of Marine Sciences, University of Haifa, 3498838, Haifa, Israel
[6] International Ocean Discovery Program, Texas A&M University, College Station, Texas 77845, U.S.A.
[7] Department of Geology and Geophysics, Texas A&M University, College Station, Texas 77843, U.S.A
[8] Instituto Oceanográfico, Universidade de São Paulo, Praça do Oceanográfico, 191, São Paulo, SP, 05508-120, Brazil
[9] Faculty of Science, Vrije Universiteit Amsterdam, 1081 HV Amsterdam, The Netherlands
[10] Institute of Geology, CEN, University of Hamburg, Bundesstraße 55, Hamburg, 20146, Germany
[11] Center for Carbonate Research, Department of Marine Geosciences, Rosenstiel School of Marine and Atmospheric Science, University of Miami, 4600 Rickenbacker Causeway, Miami, FL 33149, USA

*Correspondence to*: Montserrat Alonso-Garcia (montseag@usal.es) and Jesus Reolid (jreolid@ugr.es)

**Abstract.** Changes in sea-level are linked to glacial-interglacial variability and have been claimed as the main factor controlling the production of carbonate platform factories. The Maldives archipelago (Indian Ocean), composed of two rows of atolls that enclose an inner sea, is a very sensitive region to sea-level changes. The sediments of the Inner Sea, offer an excellent study site to explore the impact of sea-level changes on carbonate platforms. Elemental geochemical composition records, obtained by X-ray fluorescence (XRF) core-scanning, from the Maldives Inner Sea (IODP Site U1467), have been used in this work to evaluate the influence of orbitally-driven sea-level fluctuations on the carbonate production and export from the neritic environment into the Maldives Inner Sea over the last 1.3 million years. High Sr aragonite-rich carbonates (HSAC) from neritic settings are deposited in the Maldives Inner Sea during sea-level highstand intervals, increasing the values of the Sr/Ca ratio. In contrast, during sea-level lowstand periods large areas of the atolls were exposed or unable to grow and the demise in the carbonate production and sediment export is reflected as low Sr/Ca values in the Inner Sea. However, we propose that sea level is not the only factor controlling the production of HSAC during sea-level highstands since several interglacial periods before and after the Mid-Brunhes event (MBE, ~430 ka) indicate high carbonate production (high Sr/Ca). The intensity of the summer monsoon and the Indian Ocean Dipole probably modulated the production at the atolls. Marine Isotope Stage 11 stands out as a period with high sea-level and rather high carbonate production in the Maldives platform. This



extraordinary carbonate production in the Maldives atolls (and in other low latitude carbonate platforms) probably contributed to the Mid-Brunhes dissolution event through a strong shelf-to-basin fractionation of carbonate deposition.

## 1 Introduction

During the Quaternary, the Earth's climate oscillated between cold and warm periods due to variations in the planet's orbital parameters that drove the waxing and waning of the ice sheets and the establishment of glacial-interglacial cycles. During the Early Pleistocene these cycles were paced by obliquity (~41 kyr duration), whereas during the Mid-Pleistocene transition (MPT) the 100 kyr cyclicity emerged characterizing the Late Pleistocene glacial-interglacial oscillations (Imbrie et al., 1993; Raymo, 1994; Raymo et al., 2006). The drivers behind this switch in the glacial-interglacial cyclicity are still debated since

there is no change in the astronomical forcings and the effect of eccentricity on insolation patterns is too low to explain the 100 kyr cyclicity (Berger et al., 2005; Lisiecki, 2010). Therefore, the emergence of this climatic cyclicity must involve a non-linear amplification of the eccentricity cycles though different processes such as the increase in the size of ice volume (Clark et al., 2006; Abe-Ouchi et al., 2013), the influence of sea ice (Gildor and Tziperman, 2001), or the combination of several obliquity (Huybers and Wunsch, 2005) or precession cycles (Cheng et al., 2016; Hobart et al., 2023).

The MPT has been defined as a long-term interval (ca. 1250-600 kyr) in which glacial periods started to become longer, more asymmetric, and severe (Clark et al., 2006; Mcclymont et al., 2013). An important step during this transition is centered at 900 ka, when it has been proposed that Antarctic ice volume started to increase lowering sea level and allowing the ice sheets in the Northern Hemisphere for further growing (Elderfield et al., 2012; Mudelsee and Schulz, 1997; Raymo et al., 2006). The MPT also shows an outstanding switch point at ~650 ka when the North American ice sheets coalesced and the instabilities in

the Laurentide ice sheet started to produce the Hudson derived Heinrich events that characterized the following glacial periods (Hodell et al., 2008; Hodell and Channell, 2016). After the MPT, a new climatic step, the Mid-Brunhes event (MBE, centered at ~430 ka), has been defined as a shift towards warmer interglacial periods with higher sea-level and $p$CO$_2$ concentrations (Jansen et al., 1986; Lang and Wolff, 2011; Barth et al., 2018), but with a variable regional expression (Candy and Mcclymont, 2013; Mcclymont et al., 2013). The MBE has also been associated with a period of high dissolution in the deep ocean that

corresponds with an increase in coccolithophore calcification (Barker et al., 2006; Flores et al., 2003; Hodell et al., 2003), and, also, with periods of intense carbonate production in the shallow carbonate banks and platforms (Droxler et al., 1990; Zeigler et al., 2003). The continental records indicate that the MBE brought wetter and warmer interglacial periods (Candy et al., 2014; Zhisheng et al., 2015), and the loess records indicate an intensification of the Asian monsoon since Marine Isotope Stage (MIS) 13 (Sun et al., 2006). Moreover, the period between the MPT and the MBE represents a very important step in human evolution

which has been related to this climatic transition with the development of tool innovations, more social interactions and diversification of human lineages/species (Davis and Ashton, 2019; Moncel et al., 2020; Moncel et al., 2015; Galway-Witham et al., 2019; Ao et al., 2020).



The sea-level fluctuations produced by the glacial-interglacial climate variability directly affect the growth and demise of carbonate platforms because the reef environments are very sensitive to changes in the bathymetry. Carbonate accumulation

in tropical platform environments is essentially driven by sea-level, especially related to highstands (Schlager et al., 1994; Schlager, 2003). Sea-level fluctuations influence the carbonate mineralogy of the slope and basin periplatform sediments through alternately flooding and exposing the shallow banktops (Droxler et al., 1983; Boardman et al., 1986; Paul et al., 2012). Bank-derived fine sediments containing high Sr aragonite-rich carbonate (HSAC) increase in periplatform sediments during sea-level highstands, when carbonate platform flat tops are flooded and their production and export of carbonate reach their

maxima. Conversely, the production and export of those HSAC fine sediments is strongly reduced during sea-level lowstands when platform tops are exposed and karstify.

The Maldives Inner Sea (Fig. 1) sediment deposition has been shaped by the Indian Ocean currents, forced by monsoonal changes since the middle Miocene (Betzler et al., 2016a; Betzler et al., 2018; Betzler et al., 2009), similarly to other carbonate platforms (Betzler and Eberli, 2019). The South Asian Monsoon (SAM) generates in the Northern Indian Ocean seasonally

reversing winds and shifts in the ocean currents related to the migration of the Intertropical Convergence Zone (ITCZ)(Gadgil, 2018). The migration of the ITCZ occurs along with the latitudinal variation in the maximum insolation throughout the year. This combination of processes causes differential heating between land and sea that ultimately generates the monsoon dynamics (Gadgil, 2003; Wang, 2009). In general, during the summer monsoon (June to September), SW winds prevail bringing precipitation over South Asia and promoting the flow of the Southwest Monsoon Current, whereas during the winter

season (November-April), NE winds promote the flow of the North Equatorial current westwards (Tomczak and Godfrey, 2003). During the intermonsoon seasons, particularly in October-November, intense westerly winds develop in the equatorial region, the Indian Ocean Equatorial Westerlies (IEW) or Wyrtki jets (Wyrtki, 1973). Those westerly winds, introduce strong currents into the Maldives Inner Sea and produce intense mixing of the upper water column.

The general atmospheric circulation of the equatorial Indian Ocean develops a west-to-east circulation pattern, following the

sea surface temperature (SST) gradient (cooler in the western part than in the eastern region), which brings precipitation to the southeastern Indian Ocean, off Sumatra (Webster et al., 1999; Saji et al., 1999). However, this atmospheric pattern may fluctuate depending on the SST anomalies of the Indian Ocean Dipole (IOD, Fig. 2). The IOD is a coupled ocean-atmosphere phenomenon, similar to the El Niño–Southern Oscillation , which reflects changes in the east-west Indian Ocean SST gradient and in the location of rainfall (Cai et al., 2012; Saji et al., 1999). During negative IOD conditions, the east-west decreasing

SST gradient is enhanced, with cooler than normal SST in the western part of the basin and warmer than normal SST in the eastern part. This configuration leads to intense precipitation in SE Asia and Australia and intensification of the IEW, which induces strong mixing in the Maldives Sea (Hastenrath et al., 1993; Marchant et al., 2007; Saji et al., 1999). During a positive IOD phase, the normal SST gradient is reversed, with warmer than normal SST in the western part of the Indian Ocean and cooler than normal SST in the eastern part (Fig. 2). This configuration results in enhanced precipitation in East Africa and

Southern India and weak or reversed IEW.



In this study, we used elemental geochemical compositional records, obtained by X-ray fluorescence (XRF) core-scanning, from IODP Site U1467 to investigate how sea-level and coupled ocean-atmosphere dynamics affected the production and export of carbonate platform sediments to the Maldives Inner Sea over the last 1.3 Ma. Following the hypothesis of sea-level highstand shedding of carbonate platforms proposed by Schlager et al. (1994), we use the Sr/Ca ratio as a proxy for neritic

carbonate production at the Maldives platform and its export to the periplatform sediments (i.e mix of platform derived input and pelagic input). According to this hypothesis, high Sr/Ca ratios in the Maldives Inner Sea are linked to sea-level highstand conditions (high input of Sr-rich platform sediments due to the development of the carbonate factories in the atolls) and low Sr/Ca values to sea-level lowstand conditions (Droxler et al., 1990; Paul et al., 2012; Betzler et al., 2013; Boardman et al., 1986). The record of the Sr/Ca ratio has been combined with the Br normalized record, as a proxy for organic matter content

linked to pelagic primary productivity and water column mixing, and with other proxies from Site U1467 that indicate variations in the monsoon dynamics, such as the Fe/K ratio, as a proxy for summer monsoon intensity, and the Fe input for winter monsoon intensity (Kunkelova et al., 2018). The combination of all those proxies suggests that during the last 1.3 Ma changes in the carbonate production and export in the Maldives region responded to sea-level variations but also to climate fluctuations related to monsoon dynamics. Moreover, the long-term patterns observed in the records can be related to the MPT

and MBE events.

## 2 Geological setting

The Maldives archipelago (Fig. 1), situated in the central equatorial Indian Ocean, is a 3 km thick isolated tropical carbonate edifice located south-west of India (Aubert and Droxler, 1996; Purdy and Bertram, 1993). The carbonate platform was established on an early Palaeogene (60 to 50 Ma) faulted volcanic basement (Duncan and Hargraves, 1990). However,

carbonate drift deposition in the Maldives archipelago only started around 13 Ma (Betzler et al., 2009; Betzler et al., 2013; Betzler et al., 2016a; Betzler et al., 2016b; Betzler et al., 2018) coincident with a partial drowning of the Maldives carbonate platform (Betzler et al., 2009; Betzler et al., 2013; Reolid et al., 2019; Reolid et al., 2020) and the intensification of the South Asian Monsoon in this region (Betzler et al., 2016a; Betzler et al., 2018; Ling et al., 2021; Yao et al., 2023). The water depth at which the drifts were deposited changed through time from ca. 50 m at the time of the first drift deposits around 13-11 Ma

(Reolid et al., 2019) to ca. 500 m at present (Betzler et al., 2016a). The sea-level fluctuations that occurred during the Quaternary were recorded as changes in the intensity of the bioturbation (Reolid and Betzler, 2019) and as changes in the fossil assemblages in the drift sediments (Gupta et al., 2010; Bunzel et al., 2017; Sreevidya et al., 2019; Alvarez Zarikian et al., 2022).

The studied interval corresponds to the uppermost part of the last depositional drift unit (DS10 according to Betzler et al.,

2018), and comprises the last 1.3 Ma (from the seafloor to approximately 43 meters of composite depth, mcd). The sediment consists of unlithified foraminifer-rich, very fine to fine grained wackestone to packstone texture with a calcareous ooze matrix. The sediment shows periodic changes in the grain size following the glacial-interglacial climatic cycles with finer sediments during interglacial periods (Lindhorst et al., 2019; Alvarez Zarikian et al., 2022). Thick (0.3-1 m) to very thick (>1 m) intervals





defined by color changes ranging from light gray to grayish brown characterize this lithologic unit (Fig. 3A). Bioturbation is common to intense in the darker intervals and is often represented by color mottling. Burrows contain higher concentrations of particulate organic matter and commonly coarser material (Fig. 3A). Contacts between the darker and lighter intervals are gradational. Compared to the dark intervals, light packages appear more bioturbated and are depleted in particulate organic matter (Betzler et al., 2016b; Betzler et al., 2018; Reolid and Betzler, 2019). The fossil assemblage consists of abundant well-preserved planktonic foraminifers and pteropods, especially in the uppermost part of the interval (Fig. 3B). Benthic foraminifers, ostracods, mollusk fragments, echinoid fragments, and dark brown organic matter particles are present but not very abundant. In addition, cold-water coral remains were found at 7.13 m below sea floor (mbsf) and 11.90 mbsf (Fig. 3C). The finer fraction is made up of coccoliths, aragonite needles, tunicates, silicoflagellates, and sponge spicules with minor clay minerals and particulate organic matter.

## 3 Materials and Methods

The IODP Expedition 359 drilled four holes at Site U1467 in the Maldives archipelago, Indian Ocean (4° 51′ N, 73° 17′ E) at an average water depth of 487.4 m. A shipboard composite depth record was established (Betzler et al., 2017) using standard IODP procedures (Correlator software version 2.01.1). High-resolution compositing was based on sediment lightness (L*) data from Holes U1467A to U1467D, and adjusted after the cruise based on the higher resolution X-ray fluorescence (XRF) core-scanning data (Kunkelova et al., 2018). The splice data, including the splice intervals, core offsets, and tie points between each borehole, are archived on the IODP LIMS Database (http://iodp.tamu.edu/database/).

Non-destructive XRF analyses were performed at the Ocean Drilling and Sustainable Earth Science (ODASES) core scanning facility at the IODP Gulf Coast Repository (GCR) at Texas A&M University (USA), using a third-generation AvaaTech XRF scanner configured to analyze split sediment core halves for elements between Mg and U in the periodic table (Lyle and Backman, 2013). Data were acquired with a Canberra X-PIPS silicon drift detector (SDD) with 150 eV X-ray resolution at 5.9 keV and a Canberra Digital Spectrum Analyser model DAS 1000. The X-ray source was an Oxford Instruments 100 W Neptune X-ray tube with a rhodium (Rh) target. Raw spectral data were processed using the Canberra WINAXIL software package to produce elemental intensity data. The dual slit system was set to provide down-core spatial resolution of 10 mm and cross-core spatial resolution of 12 mm. The system performed two consecutive runs of the same section, the first one at 9 kV, 0.25 mA, and 6 s (for elements such as Ca, Fe, K, Al and Ti), and the second one at 30 kV, 1.25 mA and 6 s (for elements such as Sr, Br and Fe). Each core section was removed from refrigeration at least 2 h before scanning, scraped to clean and smooth the core surface, and covered with 4 μm thick Ultralene plastic film to prevent contamination of the X-ray detector. Measurements were taken at 3 cm intervals whenever possible. Some measurements were skipped or shifted to the nearest suitable area in order to avoid bad performance of the detector on cracks or uneven surfaces. To evaluate the reliability of the XRF scanning records Kunkelova et al. (2018) and (Carrasqueira et al., 2023) performed conventional XRF measurements in selected samples showing a very good match between both methodologies. To reduce the effects of water content, porosity and lithological variability (Tjallingii et al., 2007), the results of the XRF scanning have been normalized dividing the raw



total counts of each element by the total counts for each sample measured excluding Ag and Rh. When elements are not plotted in ratios they are shown as normalized, for instance Fe_n means Fe record normalized versus the total counts. In this study, following Aitchison (1982), the Sr/Ca and Fe/K ratios were normalized using Ln of the ratio, whereas the Fe/K ratio published

in Kunkelova et al. (2018) was a simple ratio between the element intensities.

Principal component analysis (PCA) and Pearson's correlation were performed using PAST (Hammer et al., 2001) with the matrix of the XRF scanning element intensities normalized. In order to identify long-term changes in the XRF records, we computed the cumulative sum (CuSum) of the deviations from the mean for Fe_n, Fe/K and Sr/Ca records. The cumulative sum method was developed for industrial control to detect changes in sequential production (Page, 1954) but it has been

extensively applied in biological oceanography (Ibanez et al., 1993; García-Comas et al., 2011). The cumulative sum of a record (i.e. a linear sequence) shows their value with respect to the long-term average. A positive slope shows a period of values greater than the long-term average, and a negative slope shows a period of values smaller than the long-term average. The steepness of the slope reflects how different a period is from the long-term average. Those changes in the tendency (that is, sequential periodical changes in the slope) can be interpreted as periodical changes in the environmental conditions related

to each proxy. For example, a change towards a drier or humid conditions from element/element XRF-scan obtained ratios (Caley et al., 2018).

The chronology of the studied interval of Site U1467 was performed correlating the Fe and K normalized records from the XRF scanning with the benthic δ¹⁸O records of ODP Sites 967–968 (Konijnendijk et al., 2015) and the Prob-stack (Ahn et al., 2017). This correlation is based on the strong response of Fe and K (as humidity proxies) to glacial-interglacial variability, and

has been previously used in Alonso-Garcia et al. (2019) and Alvarez Zarikian et al. (2022), showing an excellent agreement with the benthic δ¹⁸O age model of Stainbank et al. (2020). The age/depth table for all tie-points is included in the Appendix A.

## 4 Results

### 4.1 Geochemical Data

#### 4.1.1 Sr/Ca ratio

The Ln Sr/Ca ratio obtained from the X-ray fluorescence scanning for the study interval displays values ranging from a minimum of -0.144 and a maximum of 0.054 (Fig. 4-E). The fluctuations of this ratio show a cyclical pattern that follows the glacial-interglacial cycles observed in the benthic δ¹⁸O stack, as well as interstadial-stadial events that can be observed in the Fe-normalized record (Fig 4-B) or in the benthic δ¹⁸O record of the same site (Fig 4-A). Minimum values in the Sr/Ca ratio

occurred during cold periods whereas maximum values are associated with warm intervals. Among all the interglacial intervals, MIS 11 stands out with maximum values, followed by other interglacials such as MIS 5, MIS 1 (The Holocene), MIS 17, MIS 13 and 31. Maximum values in the Sr/Ca ratio also align well with periods of high sea-level based on the comparison with the sea-level reconstruction of ODP Site 1123 (East of New Zealand, Elderfield et al., 2012) and the sea-level stack of Spratt and





Lisiecki (2016); although in the early part of the record some of the interglacial periods present much higher sea-level than
Sr/Ca values (for example in the interval from 1000 to 900 ka). PCA and Pearson's correlation show that the Sr record is
positively correlated to Ca (r=0.71, n=2356), and negatively correlated to Fe (r=-0.97) and K (r=-0.88). Results of the Sr/Ca
ratio compared to other monsoon proxies (Kunkelova et al., 2018) are shown in figures 4 and 5, and a close up of interglacial
periods MIS 5, 11 and 13 is shown in figure 6.

### 4.1.2 Br

The Br normalized record (Br_n) obtained from the X-ray fluorescence scanning for the study interval ranges from a minimum
of 0.007 to a maximum of 0.0327 (Fig. 5-B). Br values are consistently high during glacial periods and reach their minimum
values during the optima of the interglacial periods with a sharp change from high to low values during Terminations. The
transition from interglacial to glacial periods may be gradual in the Br record but most of the transitions towards glacial periods
show an abrupt increase in Br values. The maximum value corresponds to the Last Glacial Maximum whereas the minima
values correspond to MIS 31. PCA and Pearson's correlation indicate that the Br record is positively correlated to Fe (r=0.79,
n=2356) and K (0.71), but it is negatively correlated to Ca (r=-0.71) and Sr (r=-0.85). Figure 6 shows the good coherence
between the Br and Fe records across different interglacial periods.

### 4.2 Cumulative sum analysis

The CuSum values of the U1467 Sr/Ca ratio record show three different trends over the studied interval, indicating changes in
the export and production of HSAC just at the MPT and MBE (Fig. 7). Before the MPT the HSAC production was lower than
the average; between MPT and MBE the production of HSAC was on the average; and after the MBE there is an increasing
trend that indicates the HSAC production was above the average. The CuSum values of the U1467 Fe record also show two
changes with the same timing as in the Sr/Ca record. However, in the Fe record the trend is only increasing the steepness of
the gradient, indicating higher Fe (dust) input after the MPT and even greater after the MBE (Fig. 7). The CuSum of the Fe/K
ratio record shows a similar trend to Fe, with an increase towards drier conditions after the MPT.

### 5 Discussion

### 5.1 Carbonate production and sea-level oscillations

The modern ocean is enriched in Sr mainly through riverine and hydrothermal input. Stabilization from aragonite to calcite
during sea-level lowstand periods also provides Sr to the ocean modifying the seawater Sr/Ca ratio (Stoll and Schrag, 1998).
Seawater Sr/Ca ratios are not constant globally but regionally variable (Lebrato et al., 2020); however, the mean residence
times of Sr and Ca in the ocean, ~5 and ~1 Ma respectively (Drever, 1988), allows us to interpret the U1467 Sr/Ca ratio record
as driven by regional climate-related processes. In modern carbonate platforms, aragonitic corals and green algae (such as



*Halimeda*) are the most common carbonate producers of HSAC sediments, whereas calcite producers, such as foraminifers

and coccolithophores, are more common in pelagic sediments (Schlanger, 1988; Droxler et al., 1988; Schlager et al., 1994; Betzler et al., 2013). These different biomineralization patterns determine the bulk composition of carbonate sediments in the periplatform areas. In addition, high temperature and changes in currents may also promote the abiotic formation of HSAC on the shallow platforms due to local supersaturation and may also increase the export of HSAC to the periplatform (Purkis et al., 2017; Bialik et al., 2022).

The Maldives Inner Sea has been functioning as a natural sediment trap (Belopolsky and Droxler, 2003; Betzler et al., 2009). It receives terrigenous sediments from aeolian origin (Kunkelova et al., 2018; Lindhorst et al., 2019; Yao et al., 2023; Carrasqueira et al., 2023), fragments of shallow-water organisms populating the atolls surrounding it, and planktonic and benthic pelagic microorganisms (Gupta et al., 2010; Betzler et al., 2016b; Bunzel et al., 2017; Betzler et al., 2018; Sreevidya et al., 2019; Bialik et al., 2020; Alvarez Zarikian et al., 2022). Fluctuations in the input of the different carbonate components

(mainly foraminifers and coccolithophores vs aragonitic pelagic and shallow-marine organisms), influences the Sr content of the pelagic sediments of the Maldives Inner Sea. During sea-level highstand periods, carbonate production at the platform top is at a maximum, sequestering high amounts of carbonate and producing a shelf-basin fractionation that drives dissolution in the deep basins due to the rise of the carbonate compensation depth (Hodell et al., 2001; Zeigler et al., 2003; Betzler et al., 2013). The Sr-rich carbonate that forms in the atolls is transferred to the pelagic sediments of the Maldives Inner Sea due to

storms, ocean currents and tidal erosion of the carbonate banks. During all interglacial and interstadial events, the shallow-water marine derived sediments accumulate in the pelagic environment (Fig. 8-A) increasing the U1467 Sr/Ca ratio (Fig. 4). Conversely, during times of sea-level lowstands the demise of carbonate production at the platform top, and the fact that lower sea-level prevents wave erosion of the carbonate platforms, decreases the input of sediments of platform origin (atolls and upper slope) to the pelagic deposits (Fig. 8-B). Consequently, a reduction in the Sr/Ca ratio in the periplatform sediments can

be observed (Fig. 4).

There is a clear correspondence between the sea-level records (sea-level reconstruction from off New Zealand, ODP Site 1123, by Elderfield et al., 2012 and sea-level stack by Spratt and Lisiecki, 2016) and the U1467 Sr/Ca record of the last 1.3 Ma (Fig. 4). Enhanced carbonate production and sediment export from the Maldives atolls occurred during intervals with sea-level similar or higher than the Holocene level, particularly associated with the optimum phase of interglacial/interstadial periods

(Fig. 6). Besides the uncertainties of the sea-level reconstructions and the minimal age model differences that may occur between Site U1467 and the sea-level records, it is remarkable that carbonate production and its export from the Maldives atolls (recorded as high Sr/Ca ratio in U1467) was rather active throughout most of the interglacial maxima for the last 1.3 Ma, except for the interval between MIS 29 and MIS 25 (~1030-942 ka), in which the sea surface temperatures of the Northern Indian Ocean were also slightly lower (Herbert et al., 2010). It is also remarkable that the elevation of sea-level is not

proportional to the carbonate production and sediment export. Therefore, other factors must have been involved in enhancing the production of HSAC at the carbonate platform of the Maldives. On the following sections, we will discuss other environmental controls for the production and export of HSAC.





## 5.2 Carbonate production and monsoon dynamics

The carbonate producers within the carbonate factories, including coral reef communities and other organisms living in the

atolls, are highly sensitive to sea surface temperature and to the input of nutrients (Hallock and Schlager, 1986; Hallock, 2005). Coral reefs are mostly living in warm and oligotrophic environments. The increase in nutrients stimulates plankton growth, reducing de transparency of the water, and limiting coral and calcareous algae growth. In the Maldives Sea, SST ranges from 28.0 to 29.7 °C (NOAA Coral Reef Watch site), which is optimum for reef communities, while the input of nutrients (and, as a consequence, the turbidity of the water column) depends on the wind intensity. Stronger winds promote the input of dust and

mix the upper water column, bringing nutrients to the surface. The intensity of the winds in this region is controlled by monsoon dynamics and IOD. Positive IOD phases are linked to warmer than normal SST in the western Indian Ocean and weak easterly winds (Fig. 2), which modify sea surface circulation and prolong the Indian summer monsoon season by delaying the southward movement of the ITCZ (Cai et al., 2021). Those conditions are optima for the development of coral reef communities due to the weaker winds and high SST. Sustained increases in SST may bring punctual bleaching events but coral

reefs usually recover rather quickly after them (Kench et al., 2022).

On the other hand, the negative IOD phases increase the strength of the IEW (Saji et al., 1999), which generates stronger currents, mixing the water column and upwelling nutrients. According to Hallock and Schlager (1986), these conditions produce environmental stress and reduction of coral reef growth. In addition, intensification of the monsoon winds has been related to modification of the atolls´ shorelines by erosion (Kench et al., 2005; Kench et al., 2009). Studies on present day

atolls (Kench et al., 2009; Gischler et al., 2014) have demonstrated how yearly variations in the monsoon intensity, wind and waves, modify the circulation pattern of surface currents and, accordingly, the erosional balance of the atolls. It is reasonable to suppose that, in the past, the Maldives atolls, which developed soil and vegetation as their modern counterparts, were intensely eroded by enhanced monsoonal storm-waves and wind-driven currents, modifying the morphology of the emerged areas and exporting nutrients and other elements towards the surrounding deep-environments. Indeed, the strengthening of the

ocean currents and winds has been previously claimed as a factor that generated the demise of carbonate platform production in the Maldives during the past (Betzler et al., 2009; Betzler et al., 2013).

### 5.2.1 Dust input and pelagic primary productivity

The Fe-n record of U1467 is related to the dust input from the continent and, therefore, the strength of winter monsoon. The comparison of U1467 Sr/Ca and Fe-n records shows that rather high values of Sr/Ca occurred simultaneously with low Fe

input to the Indian Ocean (Fig. 4). Fe input to the Maldives Sea increased during the glacial periods of the MPT along with the intensification of the global glacial conditions. It has been calculated that the generally colder and drier glacial conditions resulted in a twofold to fivefold increase in dust fluxes (Maher et al., 2010). The MPT aridification of the South Asian region augmented during glacial periods increasing glacial dust export (Kunkelova et al., 2018). This input of dust increases the turbidity of the water, preventing the growth of the platform top carbonate factory. In addition, the U1467 Fe/K record was





interpreted as a proxy for summer monsoon intensity by Kunkelova et al. (2018), since chemical weathering in South Asia dominates under humid conditions, leading to high Fe/K values, whereas mechanical weathering dominates under dry conditions, leading to low Fe/K values (Govin et al., 2012). The U1467 Fe/K record (Fig. 5-D) indicates an increase in chemical weathering (i.e. in humidity) in the Indian continent after each termination, with particularly humid conditions during MIS 31 and during late MIS 13. Similar alternations between dry and wet conditions in South Asia have been described in other studies,

indicating that the displacement of the ITCZ controls the precipitation in South Asia at glacial-interglacial timescales (Zhisheng et al., 2011; Ao et al., 2023; Carrasqueira et al., 2023) but also at millennial scales (Ota et al., 2022). The Sr/Ca record shows increases in the carbonate production and export at times of increases in the Fe/K ratio, indicating the influence of summer monsoon conditions in the development of the carbonate factory.

Changes in pelagic primary productivity have been inferred in this study with the Br-n record (Fig. 5-B), following previous

work in this region (Bunzel et al., 2017; Ziegler et al., 2008). The link between the monsoon intensity and surface water productivity in the Maldives Inner Sea was confirmed by Bunzel et al. (2017) who showed how low values of $\delta^{13}$C in two species of benthic foraminifera indicate enhanced vertical mixing of the water column that increased the supply of nutrients from subsurface waters into the photic zone, enhancing surface water productivity. Pelagic primary productivity was also inferred at Site U1467 using the total alkenone concentration in the sediment samples (Alonso-Garcia et al., 2019; Alvarez

Zarikian et al., 2022), although this record only reflects coccolithophore production. Both records indicate that pelagic primary productivity was rather low during the interglacial optima. We acknowledge that since the Br record is a proxy for total organic carbon (TOC) content (Ziegler et al., 2008), the high carbonate export from the platforms may be diluting the overall organic matter content. However, the Br record shows a moderate negative correlation with the Ca record indicating that carbonate dilution is not the only factor controlling the Br concentration in the sediments. The alkenone concentration record does not

follow the Ca record either. It is likely that during the glacial periods pelagic primary productivity was enhanced due to stronger winds, and probably more phyto- and zooplankton accumulated in the sediments, increasing the accumulation/preservation of organic matter. We propose that the high Br content during glacial periods is the result of higher pelagic primary productivity and lower carbonate export from the shallow-water platform (i.e. the atolls). Whereas during interglacial periods, enhanced production and sediment export of carbonate from the Maldives atolls and lower pelagic primary productivity reduced the Br

accumulation in the sediments.

Enhanced detrital input, including clays, during the glacial periods can also contribute to organic matter preservation. A significant correlation has been described between TOC preserved in the sediments and mineral surface area (mainly controlled by the presence of clay minerals) in modern and past continental margin sediments (Hedges and Keil, 1995; Kennedy et al., 2014). The role of fine terrestrial material in the production of shallow-water carbonate material is still poorly understood. In

the Bahamas, it was demonstrated that Saharan dust plays a pivotal role in the production of fine carbonate material on the platform (Swart et al., 2014). In the Maldives, the situation appears to be more complex because the Maldives are not nutrient limited during interglacial periods due to the combination of both the input of fine particles from the Bay of Bengal and upwelled waters from the west (Radice et al., 2019). In summary, the obtained Br-n record (Fig. 5) has been interpreted as a





record of organic matter accumulation, with low values corresponding to periods of low pelagic primary productivity,
concomitant with low dust input and high production and export of HSAC from the platforms.

**5.2.2 Variations of SST and monsoon intensity as control of platform sediment export**

In order to track changes in the SST of the western Indian Ocean (and thereby, possible changes in the general IOD state), we
compared the U1467 XRF scanning elemental records with the SST record of ODP Site 722 (Fig 4-F), in the Arabian Sea
(Herbert et al., 2010), which shows higher temperature variability than the U1467 SST record (Alonso-Garcia et al., 2019),
and a wider signal than the Maldives Sea. The Sr/Ca record from U1467 indicates that an elevated production and export of
HSAC in the Maldives Sea occurred at times when sea-level was high (generally similar or higher than at present), SST was
warm in the western part of the Arabian Sea (probably warmer than 26.5 ℃ at Site 722, and even warmer in the Maldives
Sea), primary productivity was low (low Br values), and summer monsoon was prolonged driving rather humid conditions in
India (high values in the Fe/K record, Fig. 5-D) and East Asia (Sun et al., 2022). Interglacial maxima seem to correspond to
periods of strong summer monsoon and persistently positive IOD conditions, which would bring warm SST and low mixing
and turbidity to the western part of the Norther Indian Ocean decreasing the pelagic primary productivity within the Maldives
Inner Sea. The lowest values of Br are recorded during those intervals, which also coincide with sea level highstands and high
HSAC export.

Small drops in sea-level usually coincide with the intensification of the winter monsoon season (Fe-n record, Fig. 4-B), and
sharp reductions in the Sr input into the basin. The strengthening of the winter monsoon may have increased the amount of
dust delivered to the Maldives augmenting the turbidity in the surface waters and hampering the growth of carbonate platform
producers along with the sea-level drop. Consequently, the carbonate export from the atolls was reduced. In addition, pervasive
neutral and negative IOD conditions probably generated strong westerly winds in the equatorial region during the inter-
monsoon season (Saji et al., 1999; Webster et al., 1999; Cai et al., 2021). This would have contributed to increase the intensity
of the sea surface currents and the mixing of the upper water column, generating more turbidity and, also, preventing the
development of coral reef communities and other carbonate platform producers (Fig. 8-C).

Summer monsoon strength and IOD conditions have been linked to the summer intertropical insolation gradient (SITIG)
between 23ºN and 23ºS on June 21st (Reichart, 1997; Mantsis et al., 2014; Bosmans et al., 2015; Alonso-Garcia et al., 2019).
The SITIG responds to the differential heating of both hemispheres at low latitudes, and modern observations suggest that this
gradient controls the tropical monsoon (Gadgil, 2003; Webster, 2004). The SITIG record (Fig. 5-F) shows a precessional
component, although it is largely controlled by obliquity because the tilt of the Earth increases/reduces the seasonal insolation
contrast between the summer and winter hemispheres, driving a higher gradient over the tropics during obliquity maxima
(Mantsis et al., 2014). High SITIG values occur when the interhemispheric pressure gradient is high, resulting in strong cross-
equatorial winds and moisture transport into the summer hemisphere, associated with an intensified winter Hadley cell in the
Southern Hemisphere and a latitudinal migration of the ITCZ towards the warmer hemisphere (Reichart, 1997; Mantsis et al.,
2014; Bosmans et al., 2015), which increases precipitation over Asia (Beck et al., 2018).


Increases in the humidity in south Asia, recorded by the U1467 Fe/K ratio, are coetaneous with high SITIG values, particularly during interglacial periods (Fig. 5). Moreover, increases in the U1467 Sr/Ca ratio also coincide with high SITIG values, which indicates that this parameter may be controlling the development of the carbonate factory through its influence on the summer

monsoon. Obviously, sea-level is the main factor controlling the production and export of HSAC in the Maldives Sea, but high SITIG and the development of positive IOD seem to favor the growth of the coral reef communities and other carbonate producers in the atolls as it can be observed in the detailed sequences of MIS 5, MIS 11 and MIS 13 (Fig. 7). The close-up of those three interglacial periods, although with probably some age uncertainty, shows that even though sea-level may be relatively high, if the SITIG is not high, dust input increases and the development of the atoll's communities is reduced,

decreasing the Sr/Ca ratio. This pattern can be observed before and after the MBE (Fig. 5 and 7), and MIS 31 presents one of the strongest SITIG gradients associated with a strong production and export of HSAC (high Sr/Ca ratio). The U1467 Fe/K ratio record suggests very humid conditions in India during MIS 31, associated with low pelagic primary productivity (low Br) and a remarkable development of the carbonate factory (Fig. 5).

### 5.3 Implications for the Mid-Pleistocene transition and Mid-Brunhes event

Together with the MPT, the MBE is considered a critical interval in the climate transitions of the Quaternary. During the MBE (ca. 430 ka) the large amplitude glacial-interglacial oscillations were established, giving rise to clear 100-kyr cycles, with severe glaciations and very warm interglacial optima that started with the MIS 12-11 climatic cycle (Jansen et al., 1986; Imbrie et al., 1993). However, some precursor events for the MBE have been described during MIS 13 and 14 (Barth et al., 2018; Ao et al., 2020). The $CO_2$ and temperature records from the Antarctic ice cores show a shift to warmer interglacial temperatures

and higher $CO_2$ levels during the interglacial periods after the MBE (Bereiter et al., 2015; Jouzel et al., 2007) but the MBE is preceded by an interval of maximum $\delta^{13}C$ values during MIS 13 (Wang et al., 2003; Wang et al., 2014). In addition, around the MBE, the Mid-Brunhes dissolution interval (MBDI) took place, with maximum dissolution centered at MIS 11 (Bassinot et al., 1994; Barker et al., 2006). The MBDI has been related to enhanced growth and calcification of the coccolitophores from the *Gephyrocapsa* complex (Rickaby et al., 2007; Gonzalez-Lanchas et al., in review) and enhanced carbonate production at

platform tops (Droxler et al., 1990; Zeigler et al., 2003), which produced a change in the ocean's alkalinity and the shallowing of the carbonate compensation depth (Droxler et al., 1990; Hodell et al., 2001).

The Indian monsoon shows a clear shift towards more intense winter monsoon at the MPT which is further enhanced at the MBE with stronger aridification during glacial periods (Kunkelova et al., 2018). This 2-step pattern can be clearly observed in the CuSum plot of the Fe record (Fig. 7-E), which shows a change at the MPT towards increasing dust input and a steeper

gradient after the MBE (even higher glacial dust input). The increase in the winter monsoon strength during glacial periods drove changes in the ocean and atmospheric patterns of the Indian Ocean, including stronger aridification in India and higher dust input to the Northern Indian Ocean (Kunkelova et al., 2018), combined with changes in mid-depth ocean ventilation due to the contraction of the oxygen minimum zone and the increase in the influence of the Antarctic Intermediate waters in the Northern Indian Ocean (Alvarez Zarikian et al., 2022). Higher bottom water ventilation at the Maldives Inner Sea is observed





during glacial periods starting at the MPT and ventilation further increased at the MBE (Alvarez Zarikian et al., 2022). This
        seafloor ventilation scenario is consistent with an intensification of the bioturbation patterns observed during interglacial
        periods (Reolid and Betzler, 2019). Other authors proposed a third step before the MBE (Ao et al., 2020), the intensification
        of the summer monsoon during MIS 13 due to the northward displacement of the ITCZ. This phenomenon created a
        temperature and precipitation asymmetry between hemispheres and a change in the global carbon cycle (Ao et al., 2020). The
ITCZ displacement is also supported by the record of the Agulhas leakage fauna, which also started to be frequent during
        interglacials since MIS 13, indicating a global change in ocean circulation during this interglacial period (Nuber et al., 2023).
        The interhemispheric asymmetry probably started with the MPT, when it has been hypothesized that the East Antarctic Ice
        sheet (EAIS) margin grew out onto the shelf driving further cooling in Antarctica and a stronger synchronicity between the
        waxing and waning of both hemispheres ice sheets (Raymo et al., 2006). The climatic change of the MBE towards globally
warmer conditions and higher sea-level during MIS 11 is probably related to the partial melt of the Greenland and West
        Antarctic Ice sheets (WAIS) as changes in the EAIS were relatively minor (Raymo and Mitrovica, 2012) or limited to MIS 5,
        9 and 11 (Wilson et al., 2018). MIS 11 may represent the initiation of interglacial periods in which both hemispheres were
        more symmetric.

        In the U1467 record, particularly outstanding values of the Sr/Ca ratio occurred during MIS 11c, indicating a period in which
410     a remarkable development of the shallow-marine carbonate factory occurred in the Maldives platform, depositing high amounts
        of Sr-rich carbonates. Similar high carbonate production has also been described for this time interval in other places such as
        the Great Barrier Reef, the Florida Keys, Belize reefs and the Great Bahama bank (Zeigler et al., 2003 and references therein).
        Moreover, it has been recently proposed that the modern tropical atolls, as we know them today, were established after the
        MBE (Droxler and Jorry, 2021). What precisely changed in the ocean to produce this change-over in the growing pattern of
415     several carbonate platforms is still being debated. It is likely that ocean circulation shifts, such as the weakening of the
        Indonesian throughflow (Petrick et al., 2019) and the intensification of the Agulhas current (Nuber et al., 2023), may have
        played an important role in modifying the monsoon system. This modification in the monsoon regime is also supported by the
        Mid-Pleistocene changes observed in the Australian Monsoon (Gong et al., 2023). MIS 11 reconstructions (Fig. 4 and 7)
        indicate that sea-level was probably the highest over the last 1 Ma (Elderfield et al., 2012; Spratt and Lisiecki, 2016; Miller et
420     al., 2020). But more importantly, the sea-level rise occurred gradually during the longest termination of the last 1 Ma (Tzedakis
        et al., 2022), providing excellent conditions for carbonate factories to thrive within the Maldives Sea and on other carbonate
        platforms. Eustatic sea-level during MIS 11 highstand (during the second half of MIS 11) has been estimated between 6 and
        13 m above present sea-level (Raymo and Mitrovica, 2012; Dutton et al., 2015). The sea-level records indicate that the eustatic
        rise was rather rapid during the termination, and after ~419 ka sea-level increase slowed down and was punctuated by small
sea-level falls (Fig. 7). It is likely that the ice sheets that disintegrated during MIS 11 (Greenland and WAIS) did not collapse
        at once but in several steps allowing for the growth of corals and other carbonate producers in the atolls. This argument would
        explain why HSAC production and export was high and pervasive during MIS 11 at the Maldives. The fact that sea-level rise
        was relatively gradual, and the SST and monsoonal conditions were optimal for carbonate platform producers triggered the





greatest carbonate production and export rates within the Maldives record. MIS 11 represents a period of very high carbonate

production and export, at least in the Maldives record, that without any doubt contributed to a change in the ocean´s alkalinity

during this interval.

However, the Maldives record shows that sea-level is not the only factor conditioning high productivity and sediment export. The interval from MIS 29 to MIS 25 is characterized by the lowest Sr/Ca values among the interglacial periods, indicating unfavorable conditions for the development of the carbonate factories even though sea-level was high. The CuSum plot for

Sr/Ca (Fig. 7-G) shows indeed a trend of decreasing carbonate production and export, particularly from MIS 31 to 22. Before the MPT, only MIS 31 and 35 show a good development of the carbonate factories, which coincides with periods of strong summer monsoon (high Fe/K ratio in Fig. 5), especially during MIS 31. Climatic records from MIS 31 show that this particularly long interglacial was characterized by warm and sustained SST (Medina-Elizalde and Lea, 2005; Weirauch et al., 2008; Herbert et al., 2010; Lawrence et al., 2010), with strong reductions in the Northern Hemisphere (NH) and West Antarctic

ice sheets (Scherer et al., 2008; Melles et al., 2012). Moreover, MIS 31 is characterized by high STIG and rather humid conditions on the continents, including strong summer monsoons (Sun et al., 2006; Girone et al., 2013; Grant et al., 2017; Oliveira et al., 2017; Justino et al., 2019). The high Sr/Ca ratio during MIS 31 coincides with high values in the Fe/K ratio, indicating that the strong summer monsoon boosted the development of the carbonate factory during this interglacial.

The Sr/Ca CuSum plot (Fig. 7-G) shows a clear change at the MPT in the HSAC production and export, from a decreasing

trend to a constant one, which suggests an increase and more constant development of the carbonate factory, with respect to the previous interval. After the MBE, the conditions changed towards stronger development of the carbonate factory during interglacial maxima. Higher HSAC production in the low latitude carbonate banks may be related to the sustained high sea-level, warm SST of the interglacial maxima and higher salinity developed in the western Indian Ocean due to a pervasive positive IOD. It can be noted that the high Sr/Ca values observed during MIS 5e, for example, follow the sea-level changes

but, also, the dust input and the SITIG oscillations (Fig. 6), which may have stimulated the carbonate deposition at the reef environments in the Maldives Archipelago. Intense summer monsoon conditions have also been inferred for the Arabian Sea during MIS 13 (Ziegler et al., 2010). According to our data (Fig. 5), the enhanced carbonate platform development during MIS 13a and 17 was clearly favored by the intensification of the summer monsoon and positive IOD, although a sea-level threshold is also necessary to provide a surface available for the development of carbonate producers in the atolls. Indeed, multiple sea-

level reconstructions indicate that the sea-level difference between those interglacials before the MBE and the Holocene was rather low (Elderfield et al., 2012; Rohling et al., 2009; Spratt and Lisiecki, 2016). MIS 17 and 13a are characterized by slightly lower global SST and sea-level, with respect to present (Medina-Elizalde and Lea, 2005; Herbert et al., 2010; Elderfield et al., 2012; Pages-Past Interglacials Working Group, 2016; Spratt and Lisiecki, 2016; Rodrigues et al., 2017), but also by very humid conditions in southern Europe and the South Asian region (Sun et al., 2006; Sánchez Goñi et al., 2019; Oliveira et al., 2020).

It has been hypothesized that the size of the NH ice sheets was probably reduced as during other interglacial periods, but the West Antarctic ice sheet was slightly larger than during the interglacials after the MBE (Shi et al., 2020). This would explain the colder temperatures recorded in Antarctic ice cores (Jouzel et al., 2007) and the lower $CO_2$ values of the "lukewarm"





interglacials compared the interglacial periods after the MBE (Bereiter et al., 2015). The slightly higher ice volume that accumulated in the West Antarctic ice sheet would have strengthened the asymmetry between both hemispheres contributing to a northward shift of the ITCZ, stronger land-sea thermal contrast, and higher precipitations in South Asia during MIS 13 and probably MIS 17 (Shi et al., 2020). As a result, all the climatic belts would shift, including the Southern Hemisphere subpolar front, which has been reported to be in a northern position before the MBE, allowing the expansion of sea ice in the Southern Ocean and reducing the primary productivity in this location (Becquey and Gersonde, 2002; Martínez-Garcia et al., 2009; Jaccard et al., 2013).

Previously, it was suggested that slightly larger NH ice sheets would have also enhanced the East Asian monsoon through the intensification of a wave train that amplifies the Asian land-ocean summer pressure gradient (Yin et al., 2009). However, this explanation for more intense Asian summer monsoon does not apply for the Indian monsoon, which is weaker if the NH ice sheets are larger (Zhisheng et al., 2011). Furthermore, many high-latitude NH localities indicate a rather warm and humid climate during those interglacial periods, incompatible with larger NH ice sheets (Prokopenko et al., 2002; Wright and Flower, 2002; De Vernal and Hillaire-Marcel, 2008; Alonso-Garcia et al., 2011a; Melles et al., 2012; Hao et al., 2015; Lozhkin et al., 2017; Barker et al., 2019; Wang et al., 2023). In addition, MIS 14 stands out in the benthic $\delta^{18}$O records as a moderate glacial period with short duration (e.g. Ahn et al., 2017) and higher sea level (Elderfield et al., 2012) than during other glacial periods (Fig. 7). The interval between MIS 15 and 13 shows very active formation of North Atlantic Deep Water (NADW) compared to other climatic cycles (Wright and Flower, 2002; Hodell et al., 2008; Alonso-Garcia et al., 2011b) and several records indicate that the rather humid and mild conditions of the interval between MIS 15 and 13 are related to the limited extent of Arctic ice sheets and the asymmetry between the Northern and Southern hemisphere ice sheets (Hao et al., 2015; Candy and Alonso-Garcia, 2018; Ao et al., 2020). In the East Asian Monsoon records MIS 15-13 is a very interesting period with weak winter monsoon and with an increasing trend in the summer monsoon proxy towards much wetter interglacial periods, being MIS 13 the first very humid interglacial period of the last 1 Ma (Ao et al., 2020; Ao et al., 2023). The Indian summer monsoon proxy (Fe/K ratio from U1467, Fig. 5) does not show the same transition towards much humid interglacials after MIS 13, but the interval MIS 15-13 was a rather humid period indicating that glacial conditions were mild during MIS 14. MIS 13 stands out as a period with a strong development of the carbonate factory for a long interval (about 35 ka), probably related to this change that affected the East Asian monsoon enhancing the rainfall during the interglacial periods. The Indian summer monsoon and carbonate factory proxies from U1467 indicate that a shift in the ocean-atmosphere dynamics, probably related to a change in the Indonesian throughflow (Petrick et al., 2019), increased the intensity of summer monsoon and favored positive IOD conditions, boosting the development of the carbonate producers in the atolls of the Maldives platform. The enhanced development of the carbonate factory also coincides with the establishment of the modern reefs (Droxler and Jorry, 2021). Further work needs to be done to unravel the exact factors that allowed this change at the MBE in the Maldives and other carbonate platforms.



## 6 Conclusions

High resolution elemental geochemical records from IODP Site U1467, in the Maldives Inner Sea, have been examined in order to assess the factors controlling carbonate production and sediment export from the atolls to the basin. The Sr/Ca ratio record indicates that most of the interglacial periods of the last 1.3 Ma presented an interval with high Sr-rich carbonate production and sediment export, which coincided with high sea-level, regionally high SST, and strong Indian summer monsoon. The U1467 records indicate that sea-level is an important factor for the development of the carbonate factory but temperature and monsoon dynamics also play a major role.

Before the MBE (~430 ka), MIS 31, 17 and 13 stand out with high Sr/Ca ratio, although the highest Sr/Ca values of the last 1.3 Ma were attained during MIS 11. After the MBE high Sr/Ca values are recorded at every interglacial optimum but they were particularly high during MIS 11, 5 and the Holocene. MIS 31, well known as a rather warm interglacial, exhibits a remarkably high development of the carbonate factory that has been explained by the sustained strong Indian summer monsoon conditions, high SST and high STIG. The high Sr/Ca ratio during MIS 17 and 13 is more surprising since those periods have been considered as lukewarm interglacials by some authors. The high-resolution elemental records of Site U1467 indicate that the Indian summer monsoon intensity and the Indian Ocean Dipole dynamics may have played a major role by reducing mixing of the upper water column (which in turn reduces turbidity) and maintaining the appropriate temperatures and salinities of sea surface waters. Regardless of the sea-level conditions the interglacial intervals between MIS 29 and 25 did not develop substantial carbonate production probably because the sea surface conditions were not optimal (i.e. IOD was probably in a negative mode and summer monsoon conditions in India were not as strong as during other interglacials). The extremely high carbonate production during MIS 11 was probably related to the gradual increase in sea-level combined with pervasive strong Indian summer monsoon conditions and positive IOD. The extensive carbonate production and sediment export at the Maldives platforms contributed to a change in the global alkalinity and to increase the carbonate dissolution during the Mid-Brunhes dissolution interval.

The Cumulative sum analysis of the U1467 proxies for Indian summer and winter monsoon indicate a 2-step increase in aridity (at the MPT and MBE) with enhanced dust input and mechanical weathering during glacial periods, probably driven by an increase in ice accumulation at both hemispheres during glacial periods. Moreover, the Cumulative sum analysis of the U1467 Sr/Ca ratio record suggests a decrease in the carbonate production and sediment export before the MPT, followed by a period with constant or moderate increase in the carbonate factory, and an intensification in the carbonate production and sediment export at the MBE. Prior to MIS 11 only MIS 31, 17 and 13 show a remarkable development of the carbonate factory in the Maldives atolls. We associate those intervals of high carbonate production with strong Indian summer monsoon conditions and positive IOD conditions coinciding with a sea-level optimum. After the MBE all the interglacial periods seem to present more favorable conditions for carbonate production in the atolls, indicating a shift in the interglacial monsoonal conditions and in the oceanographic regime.



## 7 Data availability

The data displayed in this article is available at https://zenodo.org/record/8280041.

## 8 Author contributions

MAG, JR, FJJ-E and OMB designed the study. MAG, CAZ, JR, JCL, LJ, IC, JJGR, GPE and CB were involved in performing the XRF analyses and/or funded the analyses. FJJ-E performed the statistical analyses of the data. All coauthors contributed to the interpretation and writing of the manuscript.

## 9 Competing interests

The authors declare that they have no conflict of interest.

## 535 10 Acknowledgements

We acknowledge IODP for providing the Expedition 359 sediment cores. MAG acknowledges funding from projects PICTURE (PID2021-128322NB-I00) and INDRA (EXPL/CTACLI/0612/2021). LJ e IC are supported by Fundação de Amparo a Pesquisa no Estado de São Paulo (FAPESP) process 2016/24946-9. JJGR acknowledges funding from Vrije Universiteit Amsterdam MSc- project fund. CAZ and JCL acknowledge financial support from the U.S. Science Support
Program at Columbia University under post-expedition awards OCE14–50528 and NSF award no. OCE-1326927. We also would like to thank Dick Kroon for all his work and contributions to numerous paleoceanography studies, especially those regarding to the IODP Expedition 359. He was a wonderful human being and an enthusiastic and inspiring colleague that motivated and encouraged many people, and over time we all learnt a lot from him.




**Figures and captions**

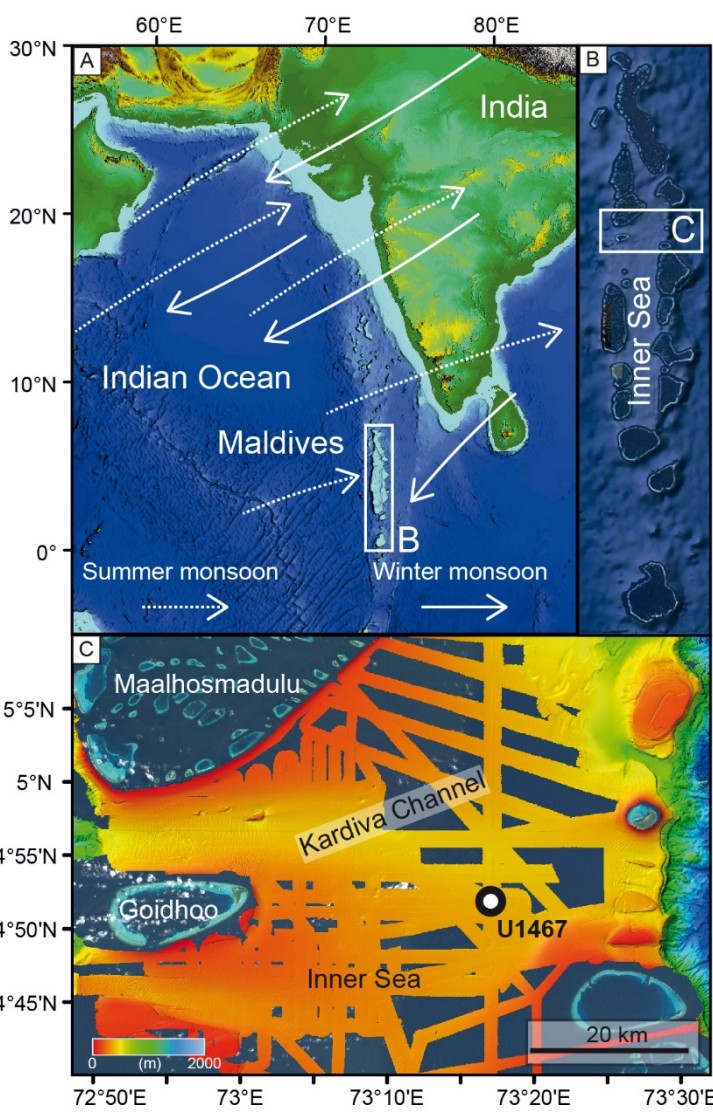


**Figure 1. A) Location of the Maldives in the Indian Ocean and representation of the wind directions for the South Asian Monsoon; B) Close up of the Maldives Archipelago showing the Inner Sea bounded by a double row of atolls; C) Close up of the study area and location of Site U1467 in the Inner Sea of the Maldives. Figure after Betzler et al. (2016a). Maps were produced using the program Esri ArcMap 10.1 (www.esri.com). Bathymetric data in A and B were exported as Geotiffs from the application**

**GeoMapApp 3.6.0 (www.geomapapp.org). In C, Worldwind satellite images (http://worldwind.arc.nasa.gov/java) were merged with multibeam data acquired during the cruises M74/4 and SO236.**






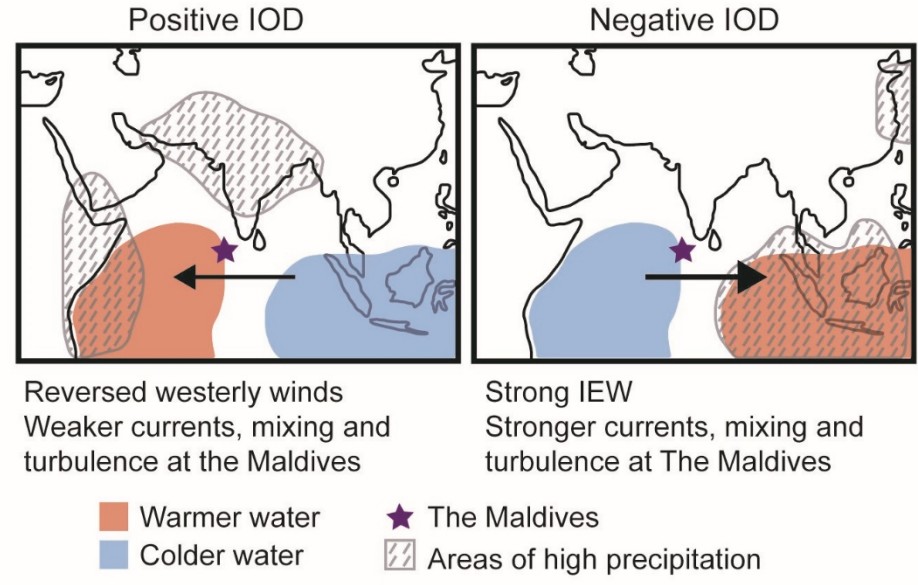

**Figure 2. Indian Ocean Dipole (IOD) modes (figure modified from Alonso-Garcia et al., 2019). The sketch shows how IOD shifts**
**precipitation in the different regions of the Northern Indian Ocean, and, also how this phenomenon affects water column mixing**
**and productivity at the Maldives Sea. Distribution of sea surface temperature and precipitation compiled from Marchant et al.**
**(2007), Hastenrath et al. (1993) and Saji et al. (1999).**





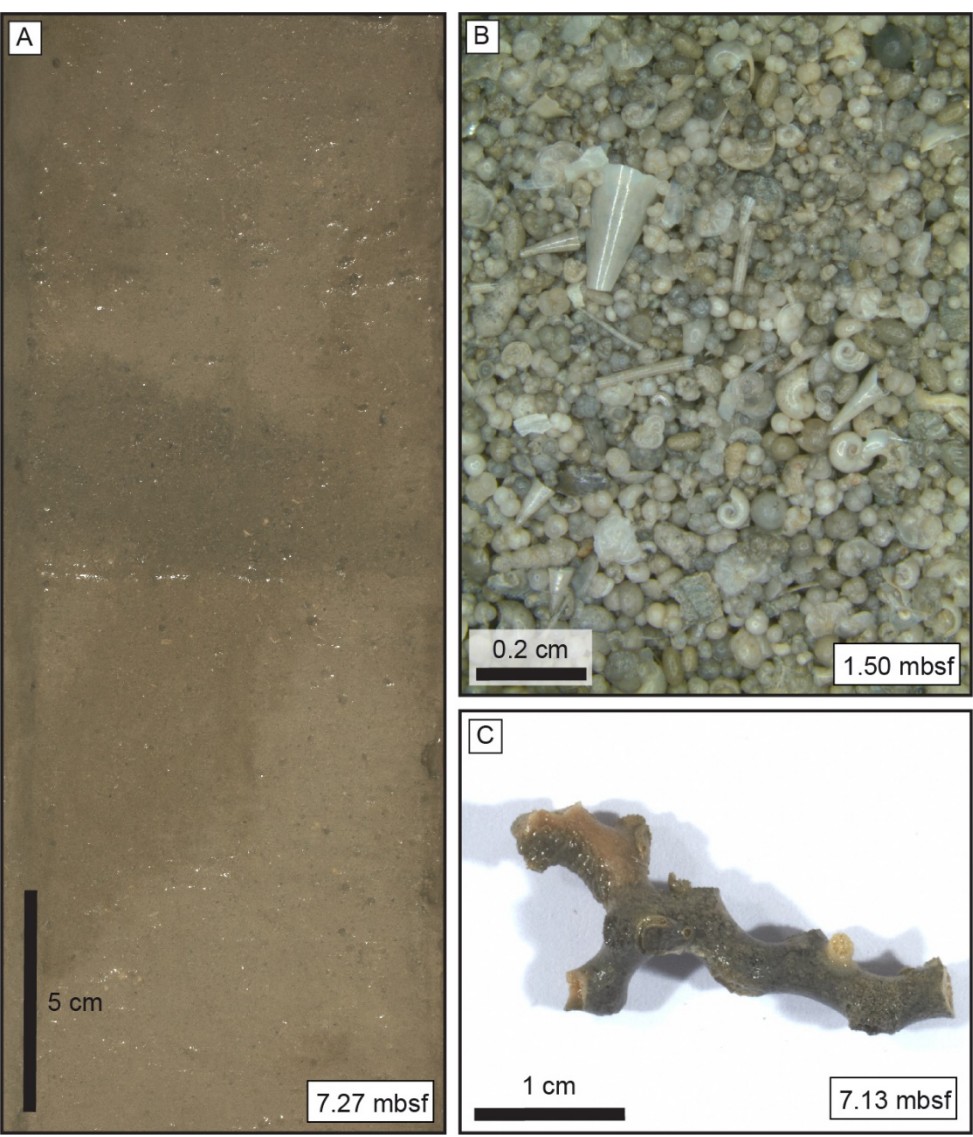

**Figure 3. Facies and bioturbation of the sheeted drift deposits at Site U1467 (modified from Reolid and Betzler, 2019). A) Core photograph of an unlithified wackestone with large trace fossils (possibly *Thalassinoides*); B) Photomicrograph of a grainstone from the uppermost part of the drift with abundant pelagic fauna including planktonic foraminifera and pteropods; C) Fragment of the cold-water coral *Lophelia Pertusa*.**



**Figure 4. XRF data from U1467 compared to other records. A) Benthic foraminifer δ¹⁸O record of Site U1467 (Stainbank et al., 2020) for stratigraphic reference; B) Fe normalized record from U1467 XRF data; C) CO₂ records for the last 1.3 Ma, light red line data from EDC3 (Bereiter et al., 2015), dark red line data from Chalk et al. (2017), and black squares data from Honisch et al. (2009); D) Sea level reconstructions from Elderfield et al. (2012) and Spratt and Lisiecki (2016); E) Sr/Ca record from U1467; F) Sea surface temperature (SST) from ODP 722 in the Arabian Sea (Herbert et al., 2010). Periods of high Sr/Ca values are highlighted by vertical gray bands except for MIS 11 which has been highlighted in orange.**

**Climate**
**of the Past**
Discussions

EGU




**Figure 5. Monsoon and productivity proxies: A) Benthic foraminifer δ¹⁸O record of Site U1467 (Stainbank et al., 2020) for stratigraphic reference; B) Br normalized record of U1467 as a proxy for primary productivity; C) Sr/Ca record from U1467 as a proxy for carbonate production and export at the Maldives platforms; D) Fe/K ratio of U1467 as a proxy for summer monsoon intensity (Kunkelova et al., 2018); E) Chinese Loess Plateau magnetic susceptibility (χ) stack, as a proxy for East Asian summer monsoon, from Ao et al. (2023); F) Summer Intertropical insolation gradient (SITIG) between 23ºN and 23ºS (Laskar et al., 2004). Periods of high Sr/Ca values are highlighted by vertical grey bands except for MIS 11 which has been highlighted in orange.**






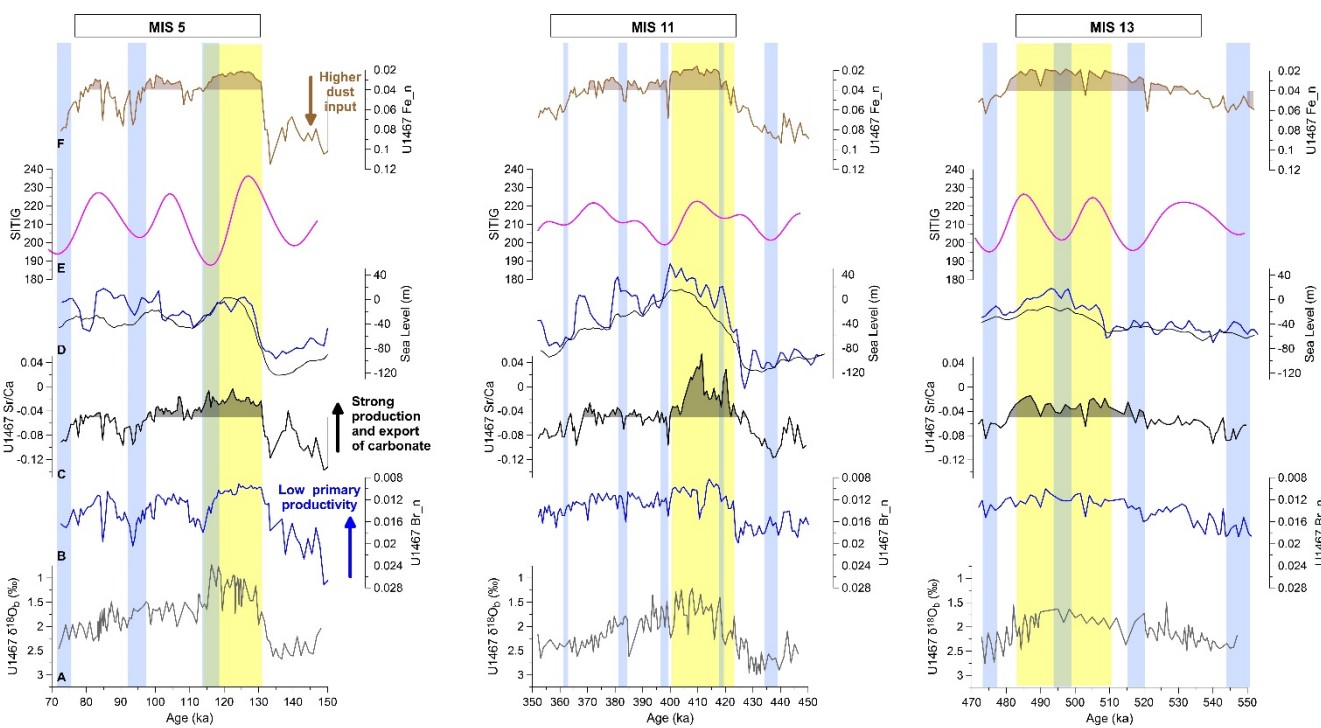


**Figure 6. Detailed sequences of the U1467 records of MIS 5, MIS 11 and MIS 13. A) Benthic foraminifer δ¹⁸O record of Site U1467**

**(Stainbank et al., 2020) for stratigraphic reference; B) Br normalized record of U1467 as a proxy for primary productivity; C) Sr/Ca record from U1467 as a proxy for carbonate production and export at the Maldives platforms; D) Sea level reconstructions from Elderfield et al. (2012) and Spratt and Lisiecki (2016); E) Summer Intertropical insolation gradient (SITIG) between 23ºN and 23ºS (Laskar et al., 2004); F) Fe normalized (Fe-n) record as a proxy for winter monsoon intensity (Kunkelova et al., 2018). Vertical blue bars indicate periods of low SITIG which are not favorable for platform growth. Vertical yellow bars indicate the main period of**

**carbonate platform production and export.**





**Figure 7. Identification of different phases in the platform growth and monsoon dynamics using the cumulative sum (CuSum) trends. A) Prob-stack benthic foraminifer δ¹⁸O record (Ahn et al., 2017) for stratigraphic reference; B) Sea level reconstructions from Elderfield et al. (2012) and Spratt and Lisiecki (2016); C) Sr/Ca record from U1467 as a proxy for carbonate production and export at the Maldives platforms; D) Sea surface temperature (SST) from ODP 722 in the Arabian Sea (Herbert et al., 2010); E) CuSum of Fe-n record; F) CuSum of Fe/K record; G) CuSum of Sr/Ca record. Green shade indicates the pre-MPT interval, Blue shade the period between MPT and MBE, and yellow shade indicates the period after MBE.**





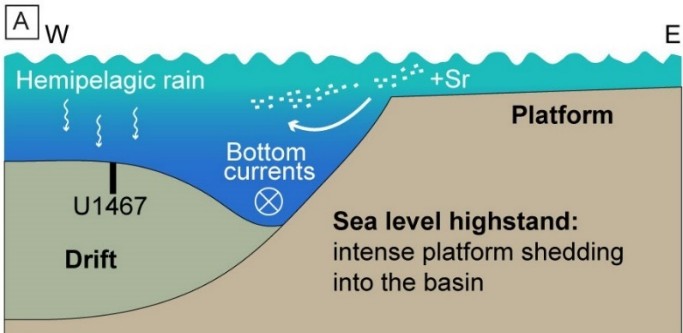

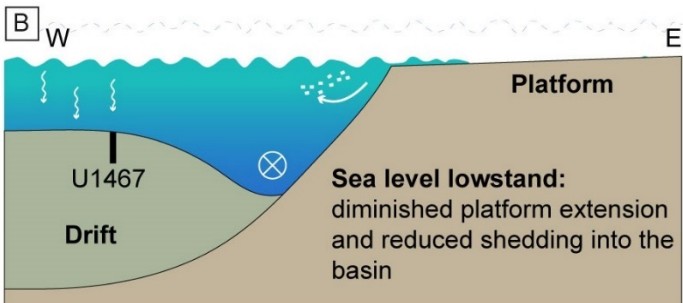

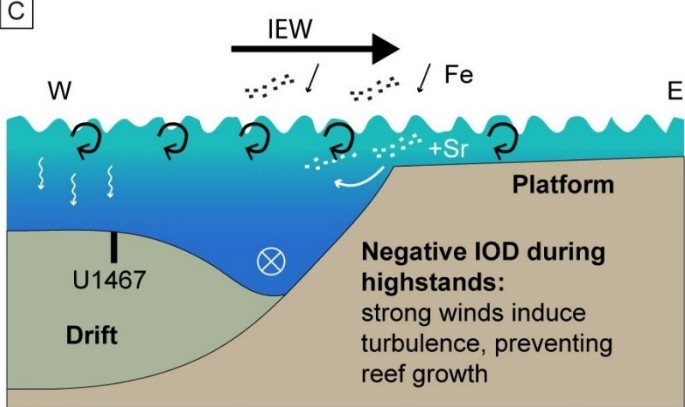

**Figure 8. Sketch depicting the main factors controlling the production of carbonate in the atolls and its export. A) Sea level highstand and optimum conditions for carbonate production at the Maldives atolls; B) Sea level lowstand drives the lowest carbonate production and export due to lower sea level; C) Carbonate production during highstands but under negative IOD that reduces carbonate production.**



## Appendix A

Age-depth control points used to reconstruct the age model.

| Depth (mcd) | Age (ka) |
|---|---|
| 0.12 | 3.40 |
| 1.08 | 15.00 |
| 2.13 | 72.00 |
| 2.85 | 89.00 |
| 4.42 | 118.38 |
| 5.185 | 133.37 |
| 6.58 | 187.00 |
| 8.89 | 219.00 |
| 9.37 | 237.00 |
| 10.18 | 246.34 |
| 11.8 | 280.34 |
| 12.35 | 290.33 |
| 12.83 | 300.33 |
| 13.82 | 317.33 |
| 14.78 | 333.00 |
| 17.4 | 400.00 |
| 18.33 | 427.30 |
| 19.74 | 474.00 |
| 20.34 | 503.00 |
| 21.33 | 540.00 |
| 22.95 | 580.00 |
| 24.79 | 620.00 |
| 27.07 | 718.00 |
| 27.7 | 754.00 |
| 29.02 | 791.21 |
| 29.62 | 814.00 |
| 31.66 | 864.00 |
| 31.84 | 872.00 |
| 33.13 | 894.00 |



| | |
|---|---|
| 33.43 | 916.00 |
| 34.09 | 958.00 |
| 35.2 | 986.00 |
| 36.01 | 1004.00 |
| 36.55 | 1034.00 |
| 37.49 | 1064.15 |
| 37.82 | 1094.00 |
| 38.3 | 1102.00 |
| 38.72 | 1120.00 |
| 39.47 | 1156.13 |
| 40.37 | 1191.12 |
| 42.11 | 1246.00 |
| 43.09 | 1288.00 |
| 44.28 | 1342.00 |








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
