# Peer review of "Sea-level and monsoonal control on the Maldives carbonate platform (Indian Ocean) over the last 1.3 million years"

_Climate of the Past, 2023_

## Author Comment (AC1)

We would like to thank the reviewer 1 (Jesse Farmer) for an insightful review with lots of great questions and comments that undoubtedly helped us to improve the manuscript.

On the following, I will answer the comments provided by the reviewer (which are listed in blue).

Review of Alonso-Garcia et al., "Sea-level and monsoonal control on the Maldives carbonate platform (Indian Ocean) over the last 1.3 million years, by Jesse Farmer

Alonso-Garcia and colleagues present x-ray fluorescence (XRF)-derived records of carbonate source (Sr/Ca), summer monsoon intensity (Fe/K), and primary productivity (Br) from a sediment record (IODP Site U1467) in the Maldives Inner Sea covering the last 1.3 million years. The authors note a first-order correlation between Sr/Ca and glacial-interglacial cycles indicative of changes in carbonate source (periplatform vs. pelagic), which they attribute to longstanding theories of how carbonate platform productivity is affected by sea level. However, they also note discrepancies between a simple sea-level driver of carbonate production. These discrepancies are highlighted for interglacials MIS 5, 11, and 13, and are attributed to a combination of summer monsoon intensity and Indian Ocean Dipole state. Last, they note long-term changes in carbonate production reaching the Maldives Inner Sea apparently coherent with the Mid-Pleistocene Transition and Mid-Brunhes Event.

Overall, this is an exciting dataset and interpretation that is worthy of publication in Climate of the Past. With that said, I think the manuscript would benefit from major revisions and another round of review. The current draft is a bit scattered; key background information appears in the discussion, and the rationale for evaluating factors other than sea level for carbonate productivity is not entirely clear. I've included a few major comments and line-by-line edits up through the discussion. I'd be happy to look at a revised manuscript and provide a more in-depth review of the discussion at that point.

 **Answer**: Thanks for your interest in our study. We will try to reorganize the text so the readers can understand better the data and discussion presented in the manuscript.

**Major comments.**

**Introduction.** As a non-expert in periplatform carbonate sedimentation, I felt that necessary background was missing in the introduction to distinguish Sr-rich carbonate production during highstands vs. Sr-poor carbonate production during lowstands. Later on, the first 1.5 paragraphs of the discussion (L224 – 245) were incredibly useful background context; I'd urge the authors to move this content from the Discussion to the Introduction.

 **Answer**: ok, I am trying to give more context in the introduction of the revised version so the readers can understand better the interpretations of the data.

**Oxygen isotope sea level proxies and their (quantitative) utility.** Whereas deep ocean d18O-based sea level reconstructions usefully indicate the glacial-interglacial character of sea level change, they are not sufficiently precise to be employed for comparing highstand sea levels:

- Typical precision on d18O-based sea level reconstructions is on the order of ± 20 m accounting solely from uncertainty on calculated d18Osw (see Ford and Raymo, 2020 -

https://doi.org/10.1130/G46546.1, and note also that the 2sigma uncertainty in the Spratt and Lisiecki, 2016 stack is similar). This magnitude of precision is not sufficient to constrain sea level differences within or between mid/late Pleistocene highstands.

- d18O-based sea level reconstructions also appear to have accuracy problems, possibly more apparent during highstands. For instance, the Elderfield Site 1123 d18O record would suggest MIS 11 sea level of + 40 m, which is hard to take seriously given geological constraints of < 13 m; Raymo & Mitrovica, 2012 (https://doi.org/10.1038/nature10891). Additionally, d18O-based sea level reconstructions appear to greatly overestimate MIS 3 sea level (Dalton et al., 2022 - https://doi.org/10.1016/j.gloplacha.2022.103814).

While such a detailed view of d18O-based sea level reconstructions might seem tangential to the current manuscript, it is necessary because the authors' motivate their investigation of additional mechanisms using the discrepancy between highstand sea level (from d18O) and Sr/Ca (L259-260). I'm not sure this avenue of motivation holds up given the imprecision on the sea level reconstructions. At the very least, the error on the sea level estimates needs to be presented. I'd also urge the authors to consider something like a crossplot of Sr/Ca vs. sea level (with uncertainty shown) to illustrate the assertion that Sr/Ca and sea level are decoupled – such decoupling is not particularly apparent in Figures 5 and 6 to warrant a long discussion of additional mechanisms.

**Answer**: This is a very good point. I will add this information about the uncertainties of the sea-level reconstructions in the text. It is especially relevant for MIS 11, when the export of HSAC is very high. Also, I will add the error of each reconstruction in Figure 4 and a cross-plot of sea level vs Sr/Ca.
The accuracy of sea-level estimates is an important point for this study, and it would be wonderful that sea-level estimates with better precision would be available. It is true that Lisiecki and Spratt (2016) acknowledged a mean uncertainty in their sea-level reconstruction of 9-12 m (1σ) for the stack. A 2σ error (18-24 m) of this record is a rather high uncertainty for places like the Maldives Archipelago. The seawater $\delta^{18}O$ based sea-level record of ODP 1123 from Elderfield et al. (2012) presented an error of ±0.2 ‰ (which is equivalent to ±20 m). However, this is one of the best sea-level estimates for the Mid-Late Pleistocene and in their supplementary material Figure S6 they showed that for the last 250 ka there was good agreement between the sea-level estimates from coral reefs and the seawater $\delta^{18}O$ derived sea-level (see Figure 1 below).

[Figure]

Figure 1. Partial reproduction of Figure S6 from Elderfield et al. (2012) from supplementary material.

The motivation of the study to find what else was driving the production and export of HSAC in the Maldives Inner Sea emerged from the differences that we can observe between the Sr/Ca record and the sea-level records, which was considered the main driver for carbonate production in the atolls and platforms. In order to answer the reviewer question, I made a cross-plot with the U1467 Sr/Ca and the Elderfield et al. (2012) sea-level record (Figure 2). The cross-plot (Figure 2A) shows a wide dispersion of the data and the linear regression a low $R^2$. Even if we only plot the values for the interglacial periods (Figure 2B) the dataset shows a wide dispersion, with very variable sea level estimates for high Sr/Ca values. This is particularly evident for the data included in the blue rectangle, which groups most of the interglacial data points. Therefore, we conclude that other factors than the sea level must be affecting the production and export of HSAC.

[Figure]

Figure 2. Cross-plot between U1467 Sr/Ca and the Elderfield et al. (2012) sea-level record showing the low correlation between both records. Panel A) shows the full dataset comparison including glacial and interglacial periods. Panel B) shows only interglacial data, green dots correspond to MIS 11 and purple dots to MIS 31. The blue rectangle indicates where most of the data is grouped.

**XRF Br_n.** Is this more reflective of productivity or diagenetic alteration? The most notable feature to me is the apparent "burn down" of Br in glacial maxima from MIS 2 to MIS 8, with effectively constant Br_n around small orbital-scale variations before this time. Could the small orbital-scale variations also be related to diagenesis?

**Answer**: Thanks for bringing up this question. We will try to clarify this point in the revised manuscript. Br is more abundant during the glacial periods in the Maldives Inner Sea (Bunzel et al., 2017 and this study), indicating higher organic carbon accumulation. According to Ziegler et al. (2008), total organic carbon (TOC) and Br content in the sediments show a clear correlation except when there is input of terrestrial organic matter. The alkenones record of U1467 does not show a strong correlation with the Br record (Figure 3), thus, the input of organic matter from the continent does not seem to be the factor controlling the Br variations. In addition, the glacial periods show higher bottom water oxygenation

based on the higher plant *n*-alcohols/*n*-alkanes (HPA) ventilation index and the ostracod assemblages (Alvarez Zarikian et al., 2022). If respiration/degradation of the organic matter would be an issue, the glacial periods would be depleted in Br and TOC but it is the opposite. Therefore, we believe the Br variations reflect sea surface productivity and that the organic matter is not very much affected by diagenetic processes. The orbital variations observed in the Br record, in our opinion, correspond to variations in sea surface productivity related to orbital parameters, mainly precession and obliquity. We will try to clarify this in the revised manuscript.

[Figure]

Figure 3. Cross-plot between U1467 Br normalized record (this study) and the U1467 alkanes concentration in the sediment (Alonso-Garcia et al., 2019).

**Minor/line-by-line comments.**

**Answer**: All the minor comments will be reformulated following the suggestions of the reviewer. Below is the answer to some questions included in the minor comments.

L31-33. This sentence confuses me. Don't you expect higher Sr/Ca during interglacial periods? If so, then "several interglacial periods before and after the Mid-Brunhes event (MBE, ~430 ka) indicate high carbonate production (high Sr/Ca)" would not be surprising. Perhaps this meant to say glacial, or some other dynamic?

**Answer**: In this sentence I meant that the interglacial periods prior to and after the MBE indicate high carbonate production. With this I mean that for example MIS 13, MIS 15, MIS 17 show similar values of Sr/Ca to MIS 9, MIS 7 or MIS 5.

References

Alonso-Garcia, M., Rodrigues, T., Abrantes, F., Padilha, M., Alvarez-Zarikian, C. A., Kunkelova, T., Wright, J. D., and Betzler, C.: Sea-surface temperature, productivity and hydrological changes in the Northern Indian Ocean (Maldives) during the interval ~575-175 ka (MIS 14 to 7), Palaeogeog. Palaeoclimatol. Palaeoecol., 536, 109376, https://doi.org/10.1016/j.palaeo.2019.109376, 2019.

Alvarez Zarikian, C. A., Nadiri, C., Alonso-García, M., Rodrigues, T., Huang, H.-H. M., Lindhorst, S., Kunkelova, T., Kroon, D., Betzler, C., and Yasuhara, M.: Ostracod response to monsoon and OMZ variability over the past 1.2 Myr, Mar. Micropaleontol., 174, 102105, https://doi.org/10.1016/j.marmicro.2022.102105, 2022.

Bunzel, D., Schmiedl, G., Lindhorst, S., Mackensen, A., Reolid, J., Romahn, S., and Betzler, C.: A multi-proxy analysis of Late Quaternary ocean and climate variability for the Maldives, Inner Sea, Clim. Past, 13, 1791-1813, 10.5194/cp-13-1791-2017, 2017.

Elderfield, H., Ferretti, P., Greaves, M., Crowhurst, S., McCave, I. N., Hodell, D., and Piotrowski, A. M.: Evolution of Ocean Temperature and Ice Volume Through the Mid-Pleistocene Climate Transition, Science, 337, 704-709, 10.1126/science.1221294, 2012.

Spratt, R. M. and Lisiecki, L. E.: A Late Pleistocene sea level stack, Clim. Past, 12, 1079-1092, 10.5194/cp-12-1079-2016, 2016.

Ziegler, M., Jilbert, T., de Lange, G. J., Lourens, L. J., and Reichart, G.-J.: Bromine counts from XRF scanning as an estimate of the marine organic carbon content of sediment cores, Geochem. Geophys. Geosyst., 9, Q05009, 10.1029/2007gc001932, 2008.

---

## Author Comment (AC2)

We would like to thank reviewer 2 (F.J. Sierro) for raising excellent questions that certainly helped us to improve the manuscript.

On the following, I will answer the comments provided by the reviewer (which are also included in blue).

Comments to the manuscript submitted by Alonso-Garcia et al.

The manuscript submitted by Alonso-Garcia and coauthors presents a very interesting record of climate variability in the Inner Sea of the Maldives archipelago along the last 1.3 My. The study is mainly based on a detailed analysis of carbonate production in the carbonate platform and export to the inner sea, which is mainly controlled by sea level and regional climate variability linked to the Indian monsoons. The manuscript is well organized and well written and deserves publication in Climate of the Past.

Below I present some suggestions that can be considered by the authors in their interpretation.

I suggest that the authors consider and discuss in the text the possibility that part of the siliciclastic particles come from river plumes. The Ganges and Brahmaputra river plumes reach today southern India to the north of Sri Lanka. These rivers generate extensive river plumes that extend along the Eastern coast of India more than 2000 km. Fournier et al., 2018 Modulation of the Ganges-Brahmaputra River Plume by the Indian Ocean Dipole and Eddies Inferred From Satellite Observations. A proxy such as Ti/Al or Zr/Al could help in this discussion.

It is clear that the variability of Sr/Ca is opposite in phase to that of Fe normalized. I suggest the authors analyze the relationship between iron contents and carbonate, by analyzing the Fe/Ca and Fe/Sr or better the Fe/(Ca+Sr) ratio. It would be important to compare the relationship between Fe/(Ca+Sr) and Fe normalized. If these two records follow the same trend, then the record of Fe normalized probably reflects the mineralogical fraction that is not calcite or aragonite and therefore the balance between carbonate and eolian supply. Because there are two main sources of carbonate supply, aragonite coming from the inner sea and the pelagic calcite source, and considering that the pelagic source is less variable, the periods with less carbonate (aragonitic) supply from the inner sea would result in higher Fe concentration in the sediments, but this does not necessarily reflect higher eolian supply, instead it could be lower aragonitic supply from the shelf. To solve this question a proxy for wind intensity could help, assuming that more intense winds can carry more eolian particles. For example, the authors can use a proxy such as Ti/Al or Ti/K, or Zr+Ti/Al+K that could reflect more the intensity of the winds and its ability to transport coarser particles to the sea. The advantage of using a proxy like this is that it is not influenced by the "closed sum" effect between carbonate and siliciclastic particles.

**Answer:** This is a very good point and indeed at the early stages of this work we considered including a proxy to evaluate riverine input but the ratios we tried were not very helpful, and the PCA analysis didn't show what could be used to discern between riverine and aeolian input. PC1 showed the differentiation between the carbonates (SR and Ca) and the detrital input (Si, K, Fe, Ti, Rb). PC2 differentiated between the carbonates rich in Ca associated with clays (rich in Fe, Si and K), and the carbonates rich in Sr mainly associated with Rb and Br. Looking at the records, it is clear that intervals rich in Sr correspond to intervals in which the input of detrital elements was reduced and carbonate production in the atolls was intense (i.e. during interglacial optima).

The problem is that the sediments from the Maldives Inner Sea are very rich in carbonate. According to the 359 Expedition report (Betzler et al., 2016), the carbonate content of Site U1467 in the upper 50 mbsf ranges from 75 to 95 wt% with an average value of 83.8 wt%. During the XRF scanning measurements we realized that Zr measurements were not reliable because the sediments are also very rich in Sr. The Sr K-beta peak (15.835 kV) overlaps the Zr K-alpha peak (15.775 kV) and, therefore, the results for Zr are not reliable in this case. Zr and Ti are elements that are included in heavy minerals such as Zircon and Rutile, respectively. It is difficult that those heavy minerals can reach the Maldives Inner Sea in a plume since they would tend to sink. Anyway, the Ti content in the XRF measurements was rather low and with many samples close to zero or even with negative numbers that indicate the element was below the detection limits (Figure 1-F).

[Figure]

Figure 1. This figure is similar to the submitted figure 5 but includes new ratios following the reviewer's suggestions. A) Benthic foraminifer $\delta^{18}O$ record of Site U1467 (Stainbank et al., 2020) for stratigraphic reference; B) Fe normalized record of U1467; C) detrital elements vs carbonate elements from U1467 showing a similar pattern to Fe_n; D) U1467 Sr/Ca; E) Fe/K ratio of U1467 as a proxy for summer monsoon intensity (Kunkelova et al., 2018); F) Ti normalized record of U1467; G) U1467 Ti/K . Periods of high Sr/Ca values are highlighted by vertical grey bands except for MIS 11 which has been highlighted in orange.

The ratios proposed by the reviewer (Ti/Al or Ti/K, or Zr+Ti/Al+K) will not be useful in Site U1467 because Ti content is very low and Zr data is not reliable. In addition, in the analyses performed by Kunkelova et al. (2018), they concluded that it was better not to use ratios with Al because of the absorption effect of XRF by pore water and water film between the sediment surface and plastic film, especially on light elements such as Al. Ti/K shows a very noisy record which slightly resembles the Fe/K record (Figure 1-G). Moreover, most of the ratios with Fe are rather similar to the Fe_n record. For example, we tried a ratio with all the detrital elements vs Ca+Sr (Figure 1-C) and the record totally resembles the Fe_n record. The same thing happens with Fe/Rb for example. Anyway, discerning between riverine and aeolian input doesn't change the discussion because the source for the detrital component would be very similar (the Indian Peninsula and nearby region). In Kunkelova et al. (2018), we demonstrated that most of the detrital elements that reached the Maldives Inner Sea are transported by the winds of the winter monsoon, which is the wind that blowns over the archipelago and can bring the dust from the Indian continent. If we have riverine input through the plumes, the source would be the same, and the elemental composition would still reflect if the region of the Indian Peninsula was affected by a wetter or drier climate.

To better understand this, it would be worth calculating Iron accumulation rates and Ca and Sr accumulation rates to have an idea about the relative contribution of the different elements.

**Answer:** The reviewer raised an interesting question, however, we do not have quantitative analyses that allow us to calculate accumulation rates.

I agree that the Fe/K ratio reflects the balance between the chemical and physical weathering and that Fe should increase during episodes of intense summer monsoon, but the supply to the Maldives inner sea must have taken place during winter monsoon that carries eolian dust from the continent to the sea. Summer monsoon winds normally flow from the ocean to the continent and do not transport dust.

**Answer:** I think, maybe this was not clear enough in the manuscript, so I will improve the text of the revised manuscript to make it clear what the Fe/K reflects. The rationale behind the interpretation of the Fe/K ratio is that if chemical weathering dominates during a time interval (for example 100 years) it is because summer monsoon was intense and, therefore, the Fe/K ratio will be high. But it doesn't mean that Site U1467 is receiving detrital sediments during the summer. As I stated abobe, Kunkelova et al. (2018), demonstrated that most of the detrital elements that reached the Maldives Inner Sea are transported by the winds of the winter monsoon, and those particles reflect the average climatic conditions, because soils do not change radically from one year to the next, or between seasons. So, even if the particles are transported by the winter monsoon winds, the composition of the detrital particles depends on the general climate over a longer period of time. If the period is dry, the Fe/K will be low, and if the period is wet the Fe/K will be high. In other words, if the summer monsoon is weak the Fe/K will be low, and vice versa.

I agree with the authors that the strong calcification event during MIS 11 is a remarkable event in the Maldives, which is also observed in the marine microplankton, and that this high surface calcification caused the Mid Brunhes dissolution event in the deep sea. This is a very important record for carbonate platforms and surface water calcification.

**Answer:** Thanks for acknowledging that this is one of the highlights of the article. The remarkable changes that took place during MIS 11 in both calcification and deep-sea dissolution are still a paradox

that is not very well resolved. Here, we tried to bring up some light into this topic, from the Maldives Sea perspective.

References

Betzler, C. G., Eberli, G. P., Alvarez-Zarikian, C. A., Alonso-Garcia, M., Bejugam, N. N., Bialik, O. M., Blättler, C. L., Guo, J. A., Haffen, S., Horozal, S., Inoue, M., Jovane, L., Kroon, D., Lanci, L., Laya, J. C., Ling, A. H. M., Lüdmann, T., Nakakuni, M., Niino, K., Petruny, L. M., Pratiwi, S. D., Reijmer, J. J. G., Reolid, J., Slagle, A. L., Sloss, C., Su, X., Swart, P. K., Wright, J. D., Yao, Z., and Young, J. R.: Expedition 359 Preliminary Report: Maldives Monsoon and Sea Level, International Ocean Discovery Program, http://dx.doi.org/10.14379/iodp.pr.359.2016, 10.14379/iodp.pr.359.2016, 2016.

Kunkelova, T., Jung, S. J. A., de Leau, E. S., Odling, N., Thomas, A. L., Betzler, C., Eberli, G. P., Alvarez-Zarikian, C. A., Alonso-García, M., Bialik, O. M., Blättler, C. L., Guo, J. A., Haffen, S., Horozal, S., Mee, A. L. H., Inoue, M., Jovane, L., Lanci, L., Laya, J. C., Lüdmann, T., Bejugam, N. N., Nakakuni, M., Niino, K., Petruny, L. M., Pratiwi, S. D., Reijmer, J. J. G., Reolid, J., Slagle, A. L., Sloss, C. R., Su, X., Swart, P. K., Wright, J. D., Yao, Z., Young, J. R., Lindhorst, S., Stainbank, S., Rueggeberg, A., Spezzaferri, S., Carrasqueira, I., Hu, S., and Kroon, D.: A two million year record of low-latitude aridity linked to continental weathering from the Maldives, Progress in Earth and Planetary Science, 5, 86, 10.1186/s40645-018-0238-x, 2018.

Stainbank, S., Spezzaferri, S., De Boever, E., Bouvier, A.-S., Chilcott, C., de Leau, E. S., Foubert, A., Kunkelova, T., Pichevin, L., Raddatz, J., Rüggeberg, A., Wright, J. D., Yu, S. M., Zhang, M., and Kroon, D.: Assessing the impact of diagenesis on foraminiferal geochemistry from a low latitude, shallow-water drift deposit, Earth Planet. Sci. Lett., 545, 116390, https://doi.org/10.1016/j.epsl.2020.116390, 2020.